# Functional Plasmon-Activated Water Increases *Akkermansia muciniphila* Abundance in Gut Microbiota to Ameliorate Inflammatory Bowel Disease

**DOI:** 10.3390/ijms231911422

**Published:** 2022-09-28

**Authors:** Chun-Chao Chang, Chih-Yi Liu, I-Chia Su, Yuarn-Jang Lee, Hsing-Jung Yeh, Wen-Chao Chen, Chih-Jui Yu, Wei-Yu Kao, Yu-Chuan Liu, Chi-Jung Huang

**Affiliations:** 1Division of Gastroenterology and Hepatology, Department of Internal Medicine, Taipei Medical University Hospital, Taipei 110, Taiwan; 2Division of Gastroenterology and Hepatology, Department of Internal Medicine, School of Medicine, College of Medicine, Taipei Medical University, Taipei 110, Taiwan; 3TMU Research Center for Digestive Medicine, Taipei Medical University, Taipei 110, Taiwan; 4Department of Pathology, Sijhih Cathay General Hospital, New Taipei 221, Taiwan; 5School of Medicine, College of Medicine, Fu Jen Catholic University, New Taipei 242, Taiwan; 6Department of Biochemistry and Molecular Cell Biology, School of Medicine, College of Medicine, Taipei Medical University, Taipei 110, Taiwan; 7Cell Physiology and Molecular Image Research Center, Wan Fang Hospital, Taipei Medical University, Taipei 110, Taiwan; 8Department of Medical Research, Cathay General Hospital, Taipei 106, Taiwan; 9Department of Biochemistry, National Defense Medical Center, Taipei 114, Taiwan

**Keywords:** plasmon-activated water, inflammatory bowel disease, *Akkermansia muciniphila*, gut microbiota, microbial biomarker

## Abstract

Inflammatory bowel disease (IBD) is associated with dysbiosis and intestinal barrier dysfunction, as indicated by epithelial hyperpermeability and high levels of mucosal-associated bacteria. Changes in gut microbiota may be correlated with IBD pathogenesis. Additionally, microbe-based treatments could mitigate clinical IBD symptoms. Plasmon-activated water (PAW) is known to have an anti-inflammatory potential. In this work, we studied the association between the anti-inflammatory ability of PAW and intestinal microbes, thereby improving IBD treatment. We examined the PAW-induced changes in the colonic immune activity and microbiota of mice by immunohistochemistry and next generation sequencing, determined whether drinking PAW can mitigate IBD induced by 2,4,6-trinitrobenzene sulfonic acid (TNBS) and dysbiosis through mice animal models. The effects of specific probiotic species on mice with TNBS-induced IBD were also investigated. Experimental results indicated that PAW could change the local inflammation in the intestinal microenvironment. Moreover, the abundance of *Akkermansia* spp. was degraded in the TNBS-treated mice but elevated in the PAW-drinking mice. Daily rectal injection of *Akkermansia muciniphila*, a potential probiotic species in *Akkermansia* spp., also improved the health of the mice. Correspondingly, both PAW consumption and increasing the intestinal abundance of *Akkermansia muciniphila* can mitigate IBD in mice. These findings indicate that increasing the abundance of *Akkermansia muciniphila* in the gut through PAW consumption or other methods may mitigate IBD in mice with clinically significant IBD.

## 1. Introduction

Inflammatory bowel disease (IBD) includes Crohn’s disease (CD) and ulcerative colitis (UC), which are chronic, relapsing–remitting systemic diseases with similar clinical burdens and treatment goals [1,2].

Dysbiosis and a dysfunctional intestinal barrier, which manifest as epithelial hyperpermeability and high levels of mucosa-associated bacteria, are observed in some patients with IBD [3,4]. Alterations in the symbiotic relationship between the host and the gut microbiota under the impact of specific environmental factors may be a major cause of IBD [4,5]. These findings indicate that the gut microbiota of patients with IBD differs greatly from that of individuals without IBD [6]. Microbe-based therapeutic approaches have been demonstrated to ameliorate IBD symptoms in both mice and humans with IBD [4,7]. Thus, manipulation of the gut microbiota may improve the treatment of IBD [8]. A better understanding of the gut microecosystem is essential to maintaining intestinal health and preventing IBD progression [9]. Plasmon-activated water (PAW), prepared by using Au nanoparticles (AuNP)-adsorbed ceramics under resonant illumination with a green light-emitting diode, has various molecular properties (e.g., antioxidant and anti-inflammatory) and functions [10,11]. Daily consumption of PAW, which has potential anti-tumor metastatic properties [11], can mitigate various inflammatory-related diseases [12,13]. However, as far as we know, few studies have focused on the anti-inflammatory properties of PAW on IBD treatment by changing the gut microbiome.

Various murine models of chronic intestinal inflammation have been developed to elucidate the pathogenetic mechanisms of IBD [14]. Dextran sodium sulfate (DSS) can induce colitis in mice [14,15], and a single intrarectal instillation of 2,4,6-trinitrobenzene sulfonic acid (TNBS) can induce severe colonic inflammation [16]. The aim of the present study was to assess the effect of PAW on anti-inflammation in mice with TNBS-induced IBD. We examined PAW-induced changes in the colonic immune activity and the gut microbiota of mice and determined whether PAW consumption was sufficient to mitigate TNBS-induced IBD and dysbiosis in mice. Herein, we showed that PAW had positive effects on anti-inflammation and the abundance of specific probiotic species in mice with TNBS-induced IBD.

## 2. Results

### 2.1. Scavenging Ability of PAW against DPPH Radicals and Response of PAW in Colonic Epithelial Cells

We first demonstrated the scavenging ability of PAW against 2,2-diphenyl-1-picrylhydrazyl (DPPH) radicals (Figure 1A). The one electron spin resonance (ESR) signal located at 3482 G is characteristic of DPPH radicals [17]. Compared to the deionized water-based solution, the respective ESR intensity significantly decreased by ca. 23% ((13,736 − 10,584) × 100%/13,736) for the PAW-based solution, as shown in ESR intensity in Figure 1B. Here, we showed that PAW caused an insignificantly histopathological change (Figure 1C, the top panel). Colon inflammation was then assessed by changes in levels of IL-6 and TNF-α, two critical proinflammatory cytokines [18,19]. The PAW-induced inflammatory response was evaluated by examining the levels of TNF-α and IL-6 in the colonic epithelial cells. As shown in Figure 1C, the colons of control mice and mice with PAW consumption for over 2 weeks showed small levels of positively stained cells of IL-6 (IHC score = 1.0, the middle panel) and TNF-α (IHC score = 1.3, the bottom panel) after the pathologists scored. In addition, no significant difference was detected in the levels of CK20 and COX-2 between the colons of normal mice that did not and did drink PAW for up to 3 months (Figure 1D). The percentages of positively stained cells of CK20 and COX-2 were all below 33% (IHC score = 1.0).

### 2.2. Induction of IBD in PAW-Drinking Mice with TNBS

An experimental mouse model of IBD involving PAW consumption and the rectal administration of TNBS was established, as shown in the study flowchart Figure 2A. Histological inflammation was assessed based on the number of epithelial neutrophils in the induced lesions using a simple scoring system: 1, normal (no inflammatory cells); 2, mild active (cryptitis but no crypt abscesses); 3, moderate active (few crypt abscesses); and 4, severe active inflammation (numerous crypt abscesses) [20]. When more than one colon biopsy sample had been collected, the highest grade of histological inflammation within that segment was recorded. The mean values of the histological scores of inflammation in the groups are presented in Figure 2B. In sum, TNBS injection resulted in severe active inflammation with architectural abnormalities, dense inflammatory cell infiltrates, and frequent crypt abscesses. Through PAW consumption, the inflammation severity was reduced with fewer architectural abnormalities and a near absence of crypt abscess (Figure 2C, the top panel). The expression of IL-6 (Figure 2C, the middle panel) and TNF-α (Figure 2C, the bottom panel) was higher (IHC score = 2.0 for IL-6 and TNF-α) in the intestinal epithelium of mice with TNBS-induced IBD, but both declined when mice further consumed PAW (IHC score = 1.0 for IL-6 and TNF-α).

### 2.3. Changes in Species Richness, Diversity, and Similarity after PAW Consumption

Alpha diversity was used to reflect the changes in gut microbiota abundance and uniformity for the groups. Figure 3 illustrates the number and diversity of bacterial species (indicating species richness and uniformity, respectively) in the gut microbiota. The stool of mice in three groups (control, TNBS, and TNBS/PAW) was subjected to gut microbiota analysis. The Chao1 index (Figure 3A) and the ACE index (Figure 3B) indicated that compared with that in the control group, the community richness of gut microbiota was substantially lower in the TNBS group but increased slightly in the TNBS/PAW group. However, the TNBS/PAW group had the lowest Shannon index scores (Figure 3C) and Simpson index scores (Figure 3D) to indicate the lowest species diversity of gut microbiota in this group.

To determine the similarities among groups of microbial communities, 3D-PCA was performed based on the correlation matrix at the ASV level and indicated by the three principal component scores of 17.3% (PC1 axis), 15.5% (PC2 axis), and 14.4% (PC3 axis), respectively (Figure 3E). Because of the high interindividual variation, clustering in the gut microbiota by TNBS-induced IBD and/or PAW intervention was evident. The results revealed that the gut microbiota of the control group and TNBS/PAW group was different from that of the TNBS group.

### 2.4. PAW Consumption Increased the Abundance of Akkermansia spp. in the Gut Microbial Community Composition

To further analyze many of the gut microbes altered by PAW, we selected the 10 most abundant microbes, accounting for 98.9% of the control group, 98.0% of the TNBS group, and 98.9% of the TNBS/PAW group, in the mouse stool on the family rank, and generated a histogram (Figure 4A). The families in each bar are sorted and displayed from bottom to top according to the overall average abundance (from high to low). The relative abundance of the *Akkermansiaceae* family in the TNBS group was considerably lower than that in the control group (7.95% versus 0.76%). In the TNBS/PAW group, the relative abundance of the *Akkermansiaceae* family was 17 times higher than that in the TNBS group. Regarding genus abundance in stool samples, the hierarchy cluster heat map revealed the 35 most abundant differentiated taxa (Figure 4B). The results demonstrated that the relative abundance of *Akkermansia* spp. was lower in the stool of the TNBS group and higher in the stool of the TNBS/PAW group. We used LEfSe, a nonparametric factorial Kruskal–Wallis sum-rank test, to determine the relative abundance of gut microbes in different groups. As presented in Figure 4C, the LEfSe (as indicated by an LDA score of >2.0) demonstrated that *Akkermansia* spp. was the dominant microbe in the stool of the TNBS/PAW group. Next, we used a bubble chart to illustrate the significance of *Akkermansia* spp. with true differences between each mouse in the control and TNBS/PAW groups (Figure 4D). The metagenomeSeq data on *Akkermansia* spp. presented the significantly higher abundance of ASVs in the control and TNBS/PAW groups (Figure 4E). Spearman correlation analysis revealed a significant negative correlation between the *Akkermansia* spp. in the stool samples with *Muribaculaceae* spp., the *Eubacterium nodatum* group, *Marvinbryantia* spp., the *Eubacterium coprostanoligenes* group, *Ruminococcus* spp., and *GCA_900066575* spp. Furthermore, the *Akkermansia* spp. was positively correlated with *Roseburia* spp., *Oscillibacter* spp., and *Lachnospiraceae_UCF_001* (Figure 5).

### 2.5. Phenotypes of A. muciniphila and PAW to IBD

Two *Akkermansia* species, *A. muciniphila* and *A. glycaniphila*, were found [21]. *A*. *muciniphila* has great probiotic potential considering its physiological benefits in various clinical scenarios [22]. In addition, *A. muciniphila* may attenuate the inflammatory responses in gastrointestinal tract by regulating the gut microbial community [23]. We therefore examined the significance of *A*. *muciniphila* to IBD. *A. muciniphila* and TNBS were rectally injected into PAW-drinking mice following a new workflow (Figure 6A). In brief, *A*. *muciniphila* (10^7^ CFUs/injection) was injected eight times into the anuses of the mice before TNBS injection. During this experimental procedure, stool samples were collected to validate the growth and proliferation in the gut through quantitative real-time polymerase chain reaction. Examination of the stool samples revealed that the abundance of *A*. *muciniphila* in the stool samples was 1.81 times greater than that in the mice not administered *A*. *muciniphila* (Figure 6B).

Changes of phenotypes in TNBS-induced mice without or with *A*. *muciniphila* injection and PAW consumption were analyzed. Compared with mice that received only TNBS induction, anal bleeding was more moderate in mice with *A*. *muciniphila* injection and PAW consumption (Figure 6C). The colon length of mice became shorter in TNBS-treated mice but was restored if mice were supplemented with *A*. *muciniphila* or PAW (Figure 6D). In addition, body weight loss was attenuated in mice with TNBS-induced IBD if mice received *A. muciniphila* injection, consumed PAW, or both (Figure 6E).

### 2.6. Mitigation of IBD in Mice That Consumed PAW and Received A. muciniphila Treatment

No histological change and obviously signal intensity of IL-6 and TNF-α in the control mice (Figure 7A). However, histological analysis revealed that *A*. *muciniphila* injection and PAW consumption alleviated the inflammatory response in the TNBS-treated mice (Figure 7B). By contrast, TNBS treatment exacerbated mucosal damage with dense infiltration of inflammatory cells, and severe architectural change (upper left panel, Figure 7C). Notably, as shown in Figure 7C, *A*. *muciniphila* administration (bottom left panel), PAW pretreatment (upper right panel), or both (bottom right panel) attenuated TNBS-induced colon damage more substantially than did TNBS treatment alone, as indicated by a reduction in epithelial loss and resolution of inflammatory response. As shown in Figure 7D, the immunostaining signals of IL-6 (IHC score = 3.5) and TNF-α (IHC score = 4.0) were the highest in the colons of TNBS-treated mice but decreased when mice drank PAW (IHC score = 2.5 for IL-6 and 3.0 for TNF-α) or were treated with *A*. *muciniphila* (IHC score = 2.3 for IL-6 and TNF-α). However, the lowest immunoactivities of IL-6 (IHC score = 1.3) and TNF-α (IHC score = 1.5) in colons were detected in the TNBS-treated mice with both PAW consumption and *A*. *muciniphila*.

## 3. Discussion

The risk of colorectal neoplasia is significantly associated with an accumulative inflammatory burden. IBD is seen worldwide as an associated immune dysfunction [24]. Anti-inflammatory therapy is one of the strategies used in IBD management [25,26]. Imbalances in host–microbiota interactions may lead to undesirable immune responses, resulting in chronic gut inflammation that in turn leads to IBD development and progression [27,28,29]. Possible disease-related differences in microbiota composition may serve as markers for patients with IBD [30]. For example, Papa et al. differentiated pediatric patients with IBD in remission and during an exacerbation of the disease by using clinical indexes based on fecal microbiota composition [31]. Fukuda et al. developed a discriminant scoring system of gut microbiota to benchmark disease activity in patients with UC [32]. These suggest that modulating gut microbiota composition can induce remission in patients with IBD [33,34].

In the present study, stool samples revealed the reduced abundance of *Akkermansia* spp. in the gut of mice with TNBS-induced IBD. This abundance returned to higher levels through PAW consumption. PAW has the notable feature of scavenging free hydroxyl and 2,2-diphenyl-1-picrylhydrazyl (DPPH) radicals [17,35,36]. Now, PAW has emerged with a biomedical potential, including the anti-inflammatory properties [37,38]. Ni et al. reported that the microbial diversity and the relative abundance of specific bacterial taxa of the gut microbiota between patients with IBD and healthy individuals were different [7]. Here, we also found that TNBS decreased the number of species (species richness) in the gut microbiota, but PAW increased that. Interestingly, the diversity of gut microbiota was low in the PAW-drinking mice. It implied that some specific bacterial taxa in stool must increase due to the PAW consumption, and these were most likely the increased *Akkermansia* spp.

Data from metagenomeSeq consistently indicated the abundance of *Akkermansia* spp. in both the control and TNBS/PAW groups. Among *Akkermansia* spp., *A*. *muciniphila*, an intestinal symbiont colonizing the mucosal layer, is a promising probiotic species [22,39] that can improve the therapeutic efficacy of chemical agents or other compounds, such as Bofutsushosan and metformin [40,41]. Its potential beneficial effect in IBD has been widely reported [42], and its administration may reduce IBD severity [43]. Increasing *A*. *muciniphila* showed the superior anti-inflammatory effect to alleviate the colitis [44]. Dong et al. reported that factors promoting *Akkermansia* spp. growth in the gut may be vital in host health maintenance [45]. Similarly, in the present study, we observed that inflammation severity was low in the mice that drank PAW. This low inflammation increased the relative abundance of *Akkermansia* spp. Supplementation with *A*. *muciniphila* in humans may improve several metabolic parameters and correlate with some human disorders [46]. Herein, we further found that PAW increased the levels of some probiotic species in the colon, particularly *Akkermansia* spp., which positively co-occurred with *Roseburia* spp. and *Oscillibacter* spp. and negatively co-occurred with *Marvinbryantia* spp. *Roseburia* spp. and *Oscillibacter* spp. are known saccharolytic, butyrate-producing bacteria in the gut [47,48]. In patients with IBD, a reduction in butyrate-producing species in the gut is the major cause of dysbiosis [48]. By contrast, the elevated abundance of *Marvinbryantia* spp. is positively associated with intestinal inflammation [49,50]. IBD may be mitigated by PAW consumption through increases in the abundance of *Akkermansia* spp., *Roseburia* spp., and *Oscillibacter* spp. and a reduction in the abundance of *Marvinbryantia* spp. Taken together, the findings suggest that *A*. *muciniphila* can ameliorate IBD severity. In other words, PAW may have antitumor potential [12,49], and *A*. *muciniphila* may help repair intestinal barriers [41,51,52,53] by promoting the secretion of colonic mucins [54] and increasing the thickness of the mucus layer [55].

Our results suggest that *Akkermansia* spp. (e.g., *A*. *muciniphila*) and PAW may decrease TNBS toxicity and have beneficial effects on the colonic cells. Their protective roles in the colon are also indicated by the higher number of surviving mice that both consumed PAW and were administered *A*. *muciniphila*. Briefly, only half of the mice without any *A*. *muciniphila* administration and PAW consumption survived on the day of the experiment, and they were lighter, had a shorter colon, and were more inflammatory. In the present study, PAW enhanced the anti-IBD function of *A*. *muciniphila*; this finding accords with that reported in previous studies specifically, that PAW has therapeutic potential for inflammatory diseases [12]. CK20 and COX-2 are critical in the development and progression of CRC [56,57,58]. Collectively, the negative results of CK20 and COX-2 and the IHC results for IL-6 and TNF-α indicate that PAW does not increase tumorigenicity. However, as reported by Bradford et al., epithelial TNF-α may promote mucosal repair during mucosal healing in IBD [59]. This suggests that PAW-induced TNF-α production may be critical in mucosal repair for intestinal epithelial integrity. Everard et al. noted that daily gavage with *A*. *muciniphila* promoted the health of experimental mice [51]. These results and the present results imply that PAW may mitigate IBD in mice by increasing the gut abundance of *A*. *muciniphila*, a potential microbial biomarker for evaluating IBD treatment outcomes.

One limitation of our study is the use of 16S rDNA NGS to identify bacterial species in the gut microbiota. Even though 16S rDNA NGS is powerful for the comprehensive analysis of gut bacteria, the specific species involved in IBD cannot be further identified from the *Akkermansia* genus. As reviewed by Muhamad Rizal et al., this limitation may be caused by the inherent low taxonomical resolution of 16S rDNA NGS and bioinformatics analysis of results [60]. Therefore, certain validation approaches are needed, such as the specific qPCR to quantify *A*. *muciniphila* in this study. Another limitation is that the present study may not have completely elucidated the precise mechanism by which PAW and *A*. *muciniphila* may cooperatively improve the colonic health or ameliorate IBD. Future studies need to evaluate the molecular mechanisms caused by PAW and *A*. *muciniphila*, which may be helpful for improving IBD.

In summary, proper drinking water can promote physical health, especially PAW consumption. The comprehensive activities of PAW make it innovatively applicable to various fields of physics, chemistry, and medicine. Here, we first demonstrate that PAW consumption can change the gut microbiota and immunoactivity in the mouse colon. By changing the microenvironment in the body, PAW increases the proportion of probiotics, e.g., *A*. *muciniphila* and other butyrate-producing bacteria in the gut. Manifestations of IBD, including the inflammatory response, were alleviated in mice with PAW consumption or/and an *A*. *muciniphila* injection.

## 4. Materials and Methods

### 4.1. Mouse Models of TNBS-Induced IBD and Stool Collection

Male BALB/c mice (6–8 weeks old) were purchased from the National Laboratory Animal Center (Taipei, Taiwan) and maintained in the Animal Research Center at Cathay General Hospital (Taipei, Taiwan), according to the regulations of the Institutional Animal Care and Use Committee of Cathay General Hospital. All animals were housed in plastic cages (3 or 4 mice/cage) under the following conditions: Humidity (50% ± 10%), light (12/12 h light/dark cycle) and temperature (23 ± 2 °C). The mice were first quarantined for 7 days before being randomly assigned by body weight into groups. First, three groups (control group: four mice without TNBS-induced IBD; TNBS group: four mice with TNBS-induced IBD; TNBS/PAW group, three mice with TNBS-induced IBD and PAW consumption) were treated according to the study flowchart (Figure 2A). Mice with TNBS-induced IBD were rectally administered 2% TNBS (FT73268; Carbosynth, Berkshire, UK) in 45% ethanol (Merck, Darmstadt, Germany) using a vinyl catheter positioned 3.5 cm proximal to the anus [61]. During the procedure, the mice were anesthetized using 2% isoflurane and N_2_ with 35% O_2_. Following catheter instillation, the animals were maintained in a vertical position for 30 s. The control mice underwent identical procedures but were instead administered with 45% ethanol dissolved in phosphate-buffered saline (PBS). Stool was collected from each cage on the day of euthanasia and snapped frozen at −80 °C prior to DNA purification.

### 4.2. Microbiota by 16S rDNA Next Generation Sequencing

Bacterial DNA from the stool was extracted using a QIAamp Fast DNA Stool Mini Kit (QIAGEN, Düsseldorf, Germany) and quantified using a NanoDrop spectrophotometer (Thermo Fisher Scientific, Waltham, MA, USA). 16S rDNA next generation sequencing (NGS) experiments were performed on an Illumina HiSeq sequencing platform (Illumina, San Diego, CA, USA) at BioTools (New Taipei, Taiwan). In brief, the variable V3-V4 16S rDNA regions were targeted for PCR amplification using the primer pair (5′-CCTACGGGNGGCWGCAG-3′ and 5′-GACTACHVGGGTATCTAATCC-3′). After NGS, QIIME 2 (v2020.11) was used to reproduce and interact with the scalable and extensible microbiome data [62], and cutPrimers was applied to accurately cut primers from reads [63]. Next, sequence screening, denoising, double-ended merging, and chimera removal were performed in sequence [64,65]. After denoising, amplicon sequence variants (ASVs) were obtained and classified using a QIIME 2 feature classifier (v2020.11) [66]. Taxonomic classification was conducted according to data retrieved from the Greengenes database (v13.8) [67]. Community richness indexes (Chao1 and abundance-based coverage estimator [ACE]) and bacterial diversity indexes (Simpson and Shannon) were generated for each group. Beta diversity was performed through principal component analysis (PCA) according to the correlation matrix and generated using the ggplot2 packages in R (v3.3.1) [68].

To understand the species distribution, following annotation, we selected the 10 families with the highest abundance and then generated a histogram presenting different taxonomic classes and their proportions. Next, the 35 most abundant genera of all sample groups were clustered using a heat map [69].

### 4.3. PAW and A. muciniphila Preparation

PAW was prepared as reported previously [13]. In brief, distilled water was passed through a glass tube filled with AuNP-adsorbed ceramic particles under resonant illumination with green light–emitting diodes (LEDs; maximum wavelength centered at 530 nm). Next, the PAW (pH 6.96, temperature 23.5 °C) was slowly collected drop by drop in glass bottles within 2 h for later use. The Bioresource Collection and Research Center (BCRC) commissioned the culture of one target gut microbe, *A*. *muciniphila*, under conditions adhering to BCRC recommendations. As indicated in the scheme to study the effects of PAW and *A*. *muciniphila* on IBD (Figure 6A), five groups of mice (control group: four mice without TNBS-induced IBD and PAW consumption; TNBS group: four mice with TNBS-induced IBD; TNBS/*A*. *muciniphila* group; four mice with TNBS-induced IBD and *A*. *muciniphila* injection; TNBS/PAW group; four mice with TNBS-induced IBD and PAW consumption; TNBS/ *A*. *muciniphila*/PAW group; four mice with TNBS-induced IBD, *A*. *muciniphila* injection, and PAW consumption) were raised. Briefly, the mice in the experimental group drank PAW or were rectally administered *A*. *muciniphila* (10^7^ CFU) before induction with 2% TNBS. The mice that were administered 2% TNBS continued to drink PAW every day. Those that were not administered 2% TNBS were rectally administered *A*. *muciniphila* every week until they were euthanized. The control animals were fed distilled water (pH 6.95, temperature 22.9 °C).

### 4.4. Inflammation Characterization of Colon Tissue in Mice with Induced IBD

After the treatments, the mice with induced IBD were euthanized, and the abdominal cavity was opened. The colon was isolated and opened longitudinally, and the inflammation status was characterized through macroscopic, histological, and immunohistochemical (IHC) analyses. In brief, gross examination of macroscopic damage in the full colon section included the assessment of rectal bleeding and colon length. Next, the isolated colons were fixed, dehydrated, and embedded in paraffin according to the standard histopathological technique, and 5-μm-thick sections were cut and transferred onto slides. Histological examination of the sections involved hematoxylin and eosin (H&E) staining, and IHC analysis involved the detection of inflammatory cytokines. In brief, the slides bearing the tissue sections were immersed in Tris-EDTA buffer (10 mM Tris base, 1 mM EDTA solution, and 0.05% Tween 20; pH 9.0) and incubated for 20 min on a hot plate (95–99 °C) or boiled and then cooled to room temperature for 20 min. The tissue was then blocked using a blocking solution (VECTASTAIN Elite ABC kit; Vector Laboratories, Burlingame, CA, USA) for 2 h. After three washes in PBS buffer, the sections were further incubated for 16 h at 4 °C with different CRC markers (for CK20, ab230524, 1:500; Abcam, Cambridge, UK; for COX-2, z2156, 1:50; Zeta, Arcadia, CA, USA) and anti-inflammatory cytokine antibodies (for TNF-α, GTX15821, 1:100; GeneTex, Irvine, CA, USA; for IL-6, ab208113, 1:100; Abcam). This was followed by 15-min incubation to block endogenous peroxidase by 0.3% H_2_O_2_, 60-min incubation to retrieve inflammatory cytokines using a biotin-labeled secondary antibody, appropriate development using peroxidase substrate solution, and counterstaining of each slide with hematoxylin. After dehydration and mounting, all slides immunostained with IL-6, TNF-α, CK20, or COX-2 antibody were examined and scored by pathologists for the percentage of cells showing specific immunostaining signals. Non-repeatable fields were selected at ×200 magnification to analyze the percentage of positively stained cells (score 1, 0–25%; score 2, 26–50%; score 3, 51–75%; score 4, 76–100%) for the final IHC score [70,71].

### 4.5. Sample Preparation for Measuring DPPH Free Radicals

DPPH was dissolved in methanol to prepare a 4 mM DPPH solution. The prepared DPPH solution and different kinds of water were then mixed (100 μL each) in a microtube. The final concentration of DPPH in the solution was 2 mM. Exactly 10 min after mixing DPPH and water, an electron spin resonance (ESR) analysis was performed. To measure an ESR spectrum, a sample was scanned once (for ca. 42 s).

### 4.6. Measurement of Free Radicals by Electron Spin Resonance Spectroscopy

For the ESR measurement, a Bruker EMX ESR spectrometer was employed. ESR spectra were recorded at room temperature using a quartz flat cell designed for solutions. The dead time between sample preparation and ESR analysis was exactly 10 min for experiments in DPPH free radicals [17] after the last addition. Conditions of ESR spectrometry were as follows: 20 mW of power at 9.78 GHz, with a scan range of 100 G and a receiver gain of 6.32 × 10^4^.

### 4.7. Statistical Analysis

The significance of between-group differences was assessed using the Student *t* test. The Kruskal–Wallis post hoc test was used for comparisons of multiple groups. Data are presented as means ± standard errors of the mean for the indicated number of independently performed experiments, and a *p* value of <0.05 was considered statistically significant. PCA was conducted to visualize the differences between the samples. The linear discriminant analysis (LDA) effect size (LEfSe) was used to determine the species with significant differences in abundance through the nonparametric factorial Kruskal–Wallis sum-rank test, and the threshold of the logarithmic LDA score for discriminative features was set to 2.0 [72]. The relative abundance of the target genera among groups was further analyzed using the R package metagenomeSeq [73]. To validate the importance of the target genera among different samples, we selected the abundant ASVs (>1%), which also appeared in at least one sample, presenting them as bubble charts [74]. To obtain the target genera–correlated microbes, we used Spearman’s test correlation analysis to explore the 30 most abundant species.

## 5. Conclusions

These findings on PAW are the first to be reported to regulate the state of the gut by changing the relevant microbiota. Increasing the abundance of *A*. *muciniphila* in the gut through PAW consumption or other methods may mitigate IBD in mice with clinical significance.

## Figures and Tables

**Figure 1 ijms-23-11422-f001:**
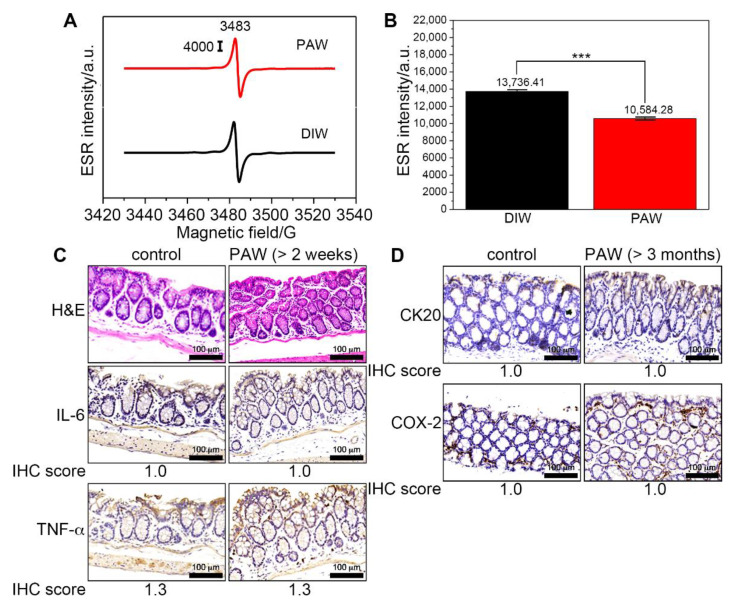
ESR experiments and histologic examinations of mice based on different kinds of water. (**A**) ESR spectra of DPPH free radicals. (**B**) The corresponding histograms showing the intensities of DPPH. (**C**) Representative images of H&E-stained and IHC staining colon sections of mice with PAW consumption for two weeks. (**D**) Representative images of IHC staining colon sections of mice with PAW consumption for CK20 and COX-2. IHC score for the percentage of positively stained cells: 1, 0–25%; 2, 26–50%; 3, 51–75%; 4, 76–100%. DIW, deionized water; PAW, plasmon-activated water; TNBS, 2,4,6-trinitrobenzene sulfonic acid; H&E, hematoxylin and eosin; IHC, immunohistochemistry. Scale bars, 100 μm. *** *p* < 0.001.

**Figure 2 ijms-23-11422-f002:**
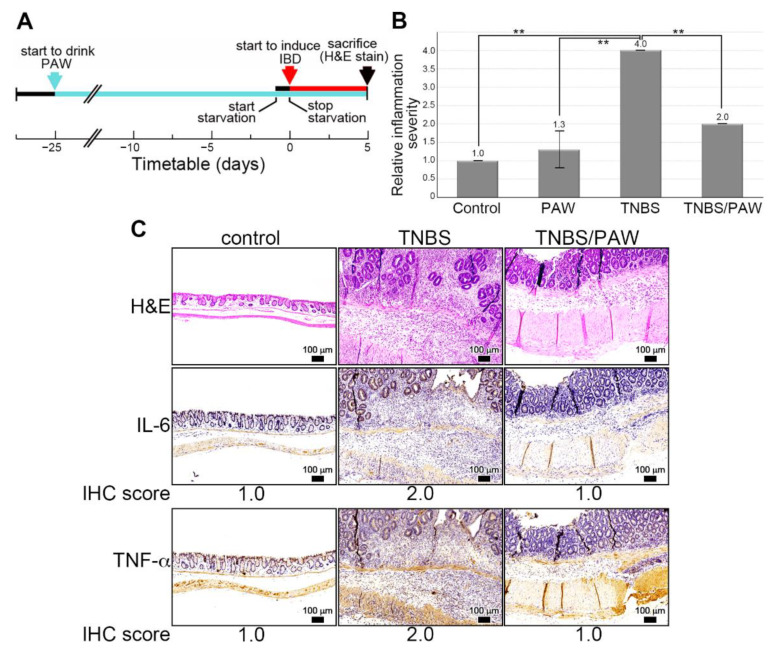
TNBS-induced IBD in mice drinking PAW. (**A**) Experimental protocol for TNBS-induced IBD and PAW consumption. The colors of the arrows denoted the time to start PAW consumption (light blue), induced IBD (red), and end experiment (black). (**B**) Histological score of inflammation in different groups. Score for inflammation severity: 1, none; 2, mild; 3, moderate; 4, severe. (**C**) Representative images of H&E-stained and IHC staining colon sections of mice with TNBS induction and PAW consumption. IL-6 and TNF-α were used as the proinflammatory cytokines. IHC score for the percentage of positively stained cells: 1, 0–25%; 2, 26–50%; 3, 51–75%; 4, 76–100%. TNBS, 2,4,6-trinitrobenzene sulfonic acid; PAW, plasmon-activated water; H&E, hematoxylin and eosin; IHC, immunohistochemistry. Scale bars, 100 μm. ** *p* < 0.01.

**Figure 3 ijms-23-11422-f003:**
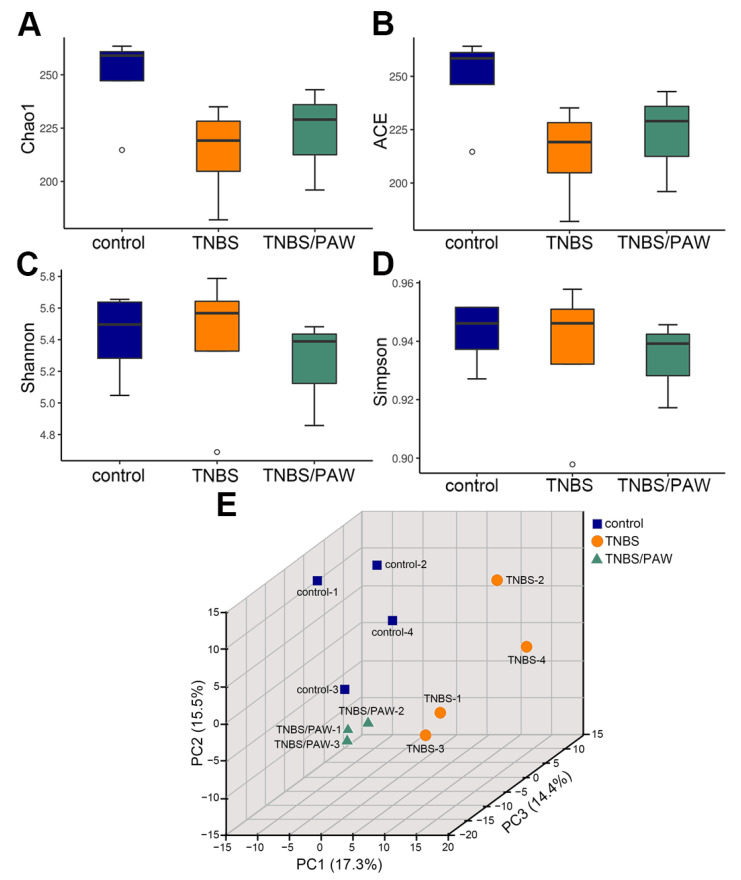
Changes in species richness, diversity, and similarity of gut microbiota in mice after TNBS induction and PAW consumption. (**A**) Chao1 and (**B**) ACE estimated the community richness for different treatments. (**C**) Shannon and (**D**) Simpson indicated the community diversity for different treatments. (**E**) Three-dimensional plot of PCA of microbial communities. The density plot on the axis was used to identify the similarity of distribution. The X-axis represented the first axis of the ordination while displaying the density of the sample’s ordination on the first axis, and the Y-axis was the second axis of the ordination while displaying the density of the sample’s ordination on the second axis. Blank circle, outliers. Points represented samples as the following indications: blue, control mice; orange, TNBS-treated mice; green, TNBS-treated mice with PAW consumption. TNBS, 2,4,6-trinitrobenzene sulfonic acid; PAW, plasmon-activated water; ACE, abundance-based coverage estimator; 3D-PCA, three-dimensional principal component analysis.

**Figure 4 ijms-23-11422-f004:**
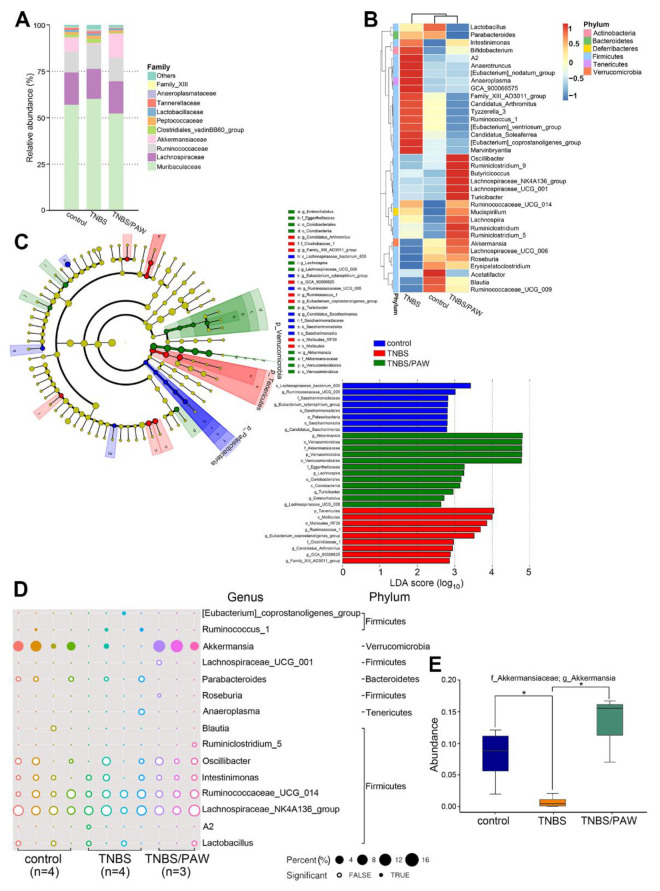
Different types of gut bacteria due to TNBS treatment and PAW consumption. (**A**) Histogram showing the relative levels of family across the three groups. (**B**) Heat map displaying the relative abundances of genus across the three groups. (**C**) Cladogram of the phylotypes that differed between the groups displayed according to effect size. Key contributors to the structural segregation of different groups were identified using LEfSe. Significant bacterial taxonomic groups are labeled, with the genus (g), family (f), species (s), or order (o). LDA scores of enriched bacterial taxa (LDA > 2 of LEfSe). Significantly enriched bacterial taxa from different groups were clustered. Differences represented by the color of the most abundant class were noted as: blue, control group; red, mice with TNBS treatment; green, mice with TNBS treatment and PAW consumption. LDA, linear discriminate analysis. (**D**) Bubble chart showing the relative abundance of amplicon sequence variants between genera (y-axis) in relation to the eleven different samples (x-axis). (**E**) MetagenomeSeq analyzing the relative abundance of the target *Akkermansia* genus among groups. The thick line in the boxplot represented the median number of reads for the cluster. * *p* < 0.05.

**Figure 5 ijms-23-11422-f005:**
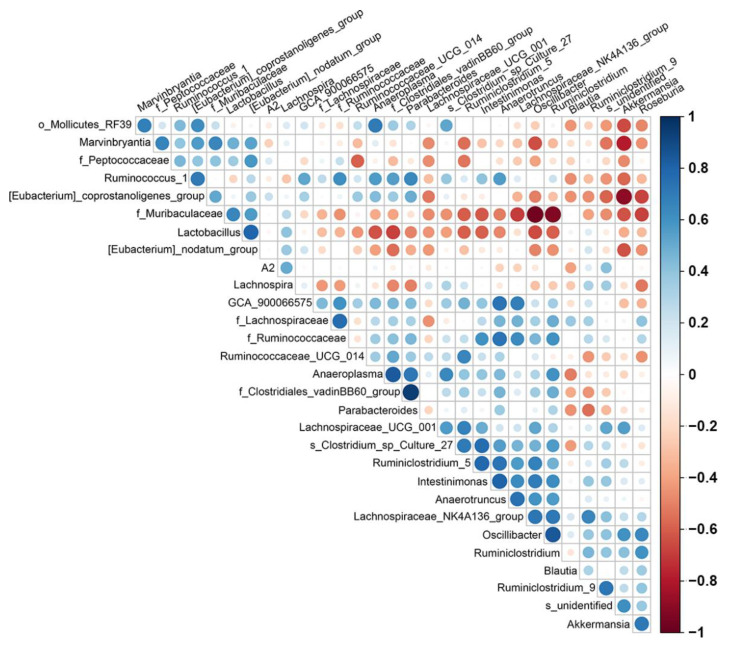
Spearman correlations between ASV-level microbial. Strong correlations are indicated by large circles, whereas weaker correlations are indicated by small circles. The colors of the circles denoted the nature of the correlation with dark blue indicating strong positive correlation and dark red indicating a strong negative correlation.

**Figure 6 ijms-23-11422-f006:**
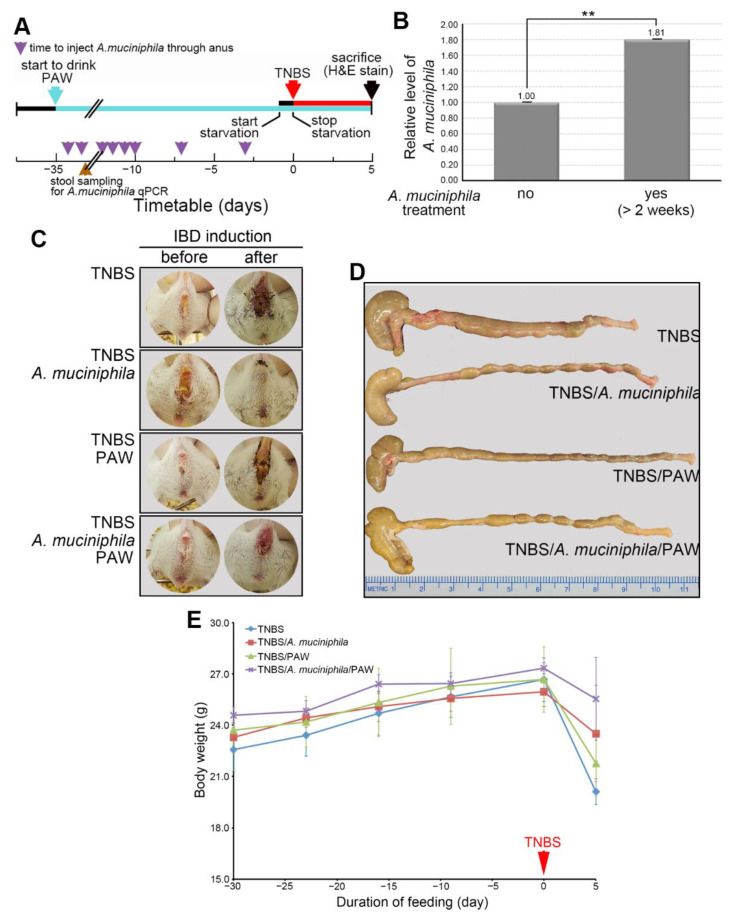
Significance of *A*. *muciniphila* to IBD. (**A**) Experimental protocol for TNBS-induced IBD, PAW consumption, and the rectal administration of *A*. *muciniphila*. The colors of the arrows denoted the time to start PAW consumption (light blue), induced IBD (red), sample stool for qPCR (brown), inject *A*. *muciniphila* through anus (purple), and end experiment (black). (**B**) Validation of *A*. *muciniphila* in stool of mice by qPCR. qPCR was performed with primers for *A*. *muciniphila* (BPID00026A, QIAGEN, Düsseldorf, Germany) and normalized with the total bacterial count (qPCR primers for total bacteria: 5′-GTGSTGCAYGGYYGTCGTCA-3′ and 5′-ACGTCRTCCMCNCCTTCCTC-3′; S for C/G, Y for C/T, R for A/G, M for A/C, and N for A/C/G/T). (**C**) Representative images of anus of mice. The anus of mice in different groups was imaged on the days before and after TNBS induction. (**D**) Representative images of colon length of mice. The bulging end of each colon is the mouse cecum. (**E**) Original body weight of mice throughout the experimental protocol. IBD, inflammatory bowel disease; TNBS, 2,4,6-trinitrobenzene sulfonic acid; PAW, plasmon-activated water; qPCR, quantitative real-time polymerase chain reaction. ** *p* < 0.01.

**Figure 7 ijms-23-11422-f007:**
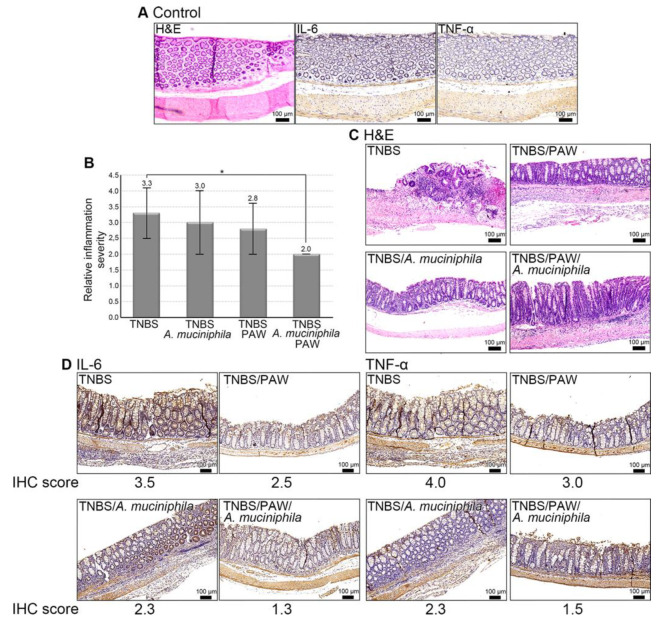
Change of inflammatory response in the presence of *A*. *muciniphila*. (**A**) Histological change and signal intensity of IL-6 and TNF-α in the control mice. Control mice, mice without PAW consumption and *A*. *muciniphila* treatment. (**B**) Histological score of inflammation in different groups. Score for inflammation severity: 1, none; 2, mild; 3, moderate; 4, severe. (**C**) Representative images of H&E-stained colon sections for inflammation severity. (**D**) Representative images of IHC staining for IL-6 and TNF-α in mouse colon. IHC score for the percentage of positively stained cells: 1, 0–25%; 2, 26–50%; 3, 51–75%; 4, 76–100%. TNBS, 2,4,6-trinitrobenzene sulfonic acid; PAW, plasmon-activated water; H&E, hematoxylin, and eosin; IHC, immunohistochemistry. Scale bars, 100 μm. * *p* < 0.05.

## Data Availability

Not applicable.

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
