# Peer review of "Functional Plasmon-Activated Water Increases *Akkermansia muciniphila* Abundance in Gut Microbiota to Ameliorate Inflammatory Bowel Disease"

_ijms, 2022, doi:10.3390/ijms231911422_

Round 1

Reviewer 1 Report

This manuscript presents an interesting investigation about the role of some gut microbiota components such as Akkermansia in inflammatory bowel disease. The manuscript is clearly drafted and the methods and results are appropriate. However I have some minor concerns that I would like to discuss before a possible recommendation to accept the manuscript. 

1. Regarding the statistical methods, analysis such as LDA should take part of the statiscal section instead of sequencing section. 

2. Which type of normalization was carried out with the raw data from microbiota sequencing? Maybe I miss this information during the reading, but in any case, the most appropiate method for normalization is centered log ratio. Please specify the normalization used in this study. 

3. The size of the figures is difficult to read, for example in figures 4 and 5.

4. Authors mentioned that beta diversity was assessed but in results section is not mentioned the results found with this analysis. 

5. Please consider to reference updated investigations, microbiota constantly advancing and some studies from 10 years ago can be consider outdated. 

6. Authors should include strengths and limitations of this investigatins. For example, 16S sequencing is not specific for the identification of bacterial species, nonetheless Akkermansia muciniphila is mentioned several times in the manuscript, which is a clear limitations of this method. 

Author Response

  1. Regarding the statistical methods, analysis such as LDA should take part of the statistical section instead of sequencing section.

Response: We thank for reviewer comment. We moved the following sentences “The linear discriminant analysis (LDA) effect size (LEfSe)…to obtain the target genera–correlated microbes, we used Spearman’s test correlation analysis to explore the 30 most abundant species.” to the section of Statistical analysis on Page 14 of the revised version.

  1. Which type of normalization was carried out with the raw data from microbiota sequencing? Maybe I miss this information during the reading, but in any case, the most appropriate method for normalization is centered log ratio. Please specify the normalization used in this study. 

Response: Here the relative abundance of each microbe in sample was defined as the ratio of individual count to the total count of microbes (Microbiome 2017 5(1):27).

  1. The size of the figures is difficult to read, for example in figures 4 and 5.

Response: We have made the all the figures in this manuscript as wide as possible.

  1. Authors mentioned that beta diversity was assessed but in results section is not mentioned the results found with this analysis.

Response: We must apologize for our omission. The revised version has been corrected.

  1. Please consider to reference updated investigations, microbiota constantly advancing and some studies from 10 years ago can be consider outdated.

Response: We appreciate reviewer’s comment. In the revised edition, the outdated references are removed, and the total number of references are reduced to 74 (including one new reference, reference #60).

  1. Authors should include strengths and limitations of this investigations. For example, 16S sequencing is not specific for the identification of bacterial species, nonetheless Akkermansia muciniphila is mentioned several times in the manuscript, which is a clear limitation of this method.

Response: Thanks for reviewer’s suggestion. As reviewer said, 16S NGS is not an absolutely good tool to identify all the bacterial species, at least not in Akkermansia genus. This really but briefly bothered us. Fortunately, only two Akkermansia species were found, as we stated in the section of Results (2.5 Phenotypes of A. muciniphila and PAW to IBD). Among these two species, A. muciniphila has been known to be one of the 2nd generation probiotics. We therefore examined the significance of A. muciniphila to IBD.

To the 5th paragraph of the section of Discussion, we have added the following sentences: One limitation of our study is the use of 16S rDNA NGS to identify bacterial species in the gut microbiota. Even though 16S rDNA NGS is powerful for the comprehensive analysis of gut bacteria, the specific species involved in IBD cannot be further identified from Akkermansia genus. As reviewed by Muhamad Rizal et al., this limitation may be caused by the inherent low taxonomical resolution of 16S rDNA NGS and bioinformatics analysis of results (reference #60, in the revised version). Therefore, certain validation approaches are needed, such as the specific qPCR to quantify A. muciniphila in this study. Another limitation is that the present study may not have completely elucidated the precise mechanism by which PAW and A. muciniphila may cooperatively improve the colonic health or ameliorate IBD. Future studies need to evaluate the molecular mechanisms caused by PAW and A. muciniphila, which may be helpful for improving IBD.

Reviewer 2 Report

The authors are to be congratulated on painstaking research in an animal model of inflammatory bowel disease that functional plasmon activated water increases Akkermansia muciniphila in the gut  and diminishes inflammation.Furthermore thy show the potential benefit  of Akkermansia muciniphila as a probiotic in reducing inflammation and that the two agents together might be potential treatments

Some very small points Line 58 using Au nanoparticles needs more explanation

Line 82 ca23 per cent needs explanation

The term infliction of IBD should be changed to TNB induced colitis  in FIG 2 and in the text and in Fig 2 There is a debate if TNB colitis is a good animal model for IBD

Line 396 a reference is given but another descriptive sentence would be helpful

Author Response

  1. Some very small points Line 58 using Au nanoparticles needs more explanation

Response: Thanks for reviewer’s carefulness. In fact, the Au nanoparticles are featured in this article. Before prepare PAW, the Au nanoparticles need to be adsorbed on ceramics first. Thus, we made a minor modification here from “…Au nanoparticles under resonant illumination…” to “…Au nanoparticles (AuNP)-absorbed ceramics under resonant illumination…” to correct the process and materials of PAW preparation. This can be also noted in the section 4.3 (PAW and A. muciniphila preparation).

  1. Line 82 ca23 percent needs explanation

Response: We have changed the related narrative to “…, the respective ESR intensity significantly decreased by ca. 23% ((13736-10584)*100%/13736) for the PAW-based solution, as shown in ESR intensity of Figure 1B” in the revised version.

  1. The term infliction of IBD should be changed to TNB induced colitis in FIG 2 and in the text and in Fig 2 There is a debate if TNB colitis is a good animal model for IBD

Response: (1) The title of Figure 2 has been changed to “TNBS-induced IBD on mice drinking PAW”. (2) We know the reviewers concerned whether TNBS-induced mouse IBD is a good animal model for studying IBD. According to our data (Figure 2B), we stated that TNBS injection resulted in severe active inflammation because of the highest histological scores (architectural abnormalities, dense inflammatory cell infiltrates, and frequent crypt abscesses) of inflammation. In addition, we have also introduced “TNBS can induce severe colonic inflammation” in the section of “Introduction”, as reviewed by Low et al. (Reference #16 in the revised version).

  1. Line 396 a reference is given but another descriptive sentence would be helpful

Response: We thank for reviewer’s comment. Indeed, the current narrative is prone to misunderstanding. Thus, we change our statement to “The PAW (pH 6.96, temperature 23.5 °C) was slowly collected drop by drop in glass bottles within 2 h for later use”.